# Robust RNA-based in situ mutation detection delineates colorectal cancer subclonal evolution

Ann-Marie Baker[1], Weini Huang[1], Xiao-Ming Mindy Wang[2], Marnix Jansen[3,4], Xiao-Jun Ma[2], Jeffrey Kim[2], Courtney M. Anderson[2], Xingyong Wu[2], Liuliu Pan[2], Nan Su[2], Yuling Luo[2], Enric Domingo[5], Timon Heide[6], Andrea Sottoriva [6], Annabelle Lewis[7], Andrew D. Beggs [8], Nicholas A. Wright[1], Manuel Rodriguez-Justo[3], Emily Park[2], Ian Tomlinson [9] & Trevor A. Graham [1]

Intra-tumor heterogeneity (ITH) is a major underlying cause of therapy resistance and disease recurrence, and is a read-out of tumor growth. Current genetic ITH analysis methods do not preserve spatial context and may not detect rare subclones. Here, we address these shortfalls by developing and validating BaseScope—a novel mutation-specific RNA in situ hybridization assay. We target common point mutations in the *BRAF*, *KRAS* and *PIK3CA* oncogenes in archival colorectal cancer samples to precisely map the spatial and morphological context of mutant subclones. Computational modeling suggests that subclones must arise sufficiently early, or carry a considerable fitness advantage, to form large or spatially disparate subclones. Examples of putative treatment-resistant cells isolated in small topographical areas are observed. The BaseScope assay represents a significant technical advance for in situ mutation detection that provides new insight into tumor evolution, and could have ramifications for selecting patients for treatment.

[1] Barts Cancer Institute, Barts and the London School of Medicine and Dentistry, Queen Mary University of London, London EC1M 6BQ, UK. [2] Advanced Cell Diagnostics, Newark, CA 94560, USA. [3] Department of Histopathology, University College London Hospital, London WC1E 6JJ, UK. [4] UCL Cancer Institute, University College London, London WC1E 6DD, UK. [5] Department of Oncology, Old Road Campus Research Building, University of Oxford, Roosevelt Drive, Oxford OX3 7DQ, UK. [6] Centre for Evolution and Cancer, The Institute of Cancer Research, 15 Cotswold Road, Sutton, London SM2 5NG, UK. [7] Cancer Gene Regulation Laboratory, Centre for Cancer Gene Research, Wellcome Trust Centre for Human Genetics, University of Oxford, Roosevelt Drive, Oxford OX3 7BN, UK. [8] Surgical Research Laboratory, Institute of Cancer and Genomic Sciences, University of Birmingham, Birmingham B15 2TT, UK. [9] Cancer Genetics and Evolution Laboratory, Institute of Cancer and Genomic Sciences, University of Birmingham, Birmingham B15 2TT, UK. Correspondence and requests for materials should be addressed to A.-M.B. (email: a.m.c.baker@qmul.ac.uk) or to I.T. (email: i.tomlinson@bham.ac.uk) or to T.A.G. (email: t.graham@qmul.ac.uk)

Recent years have seen numerous studies reveal intra-tumor heterogeneity (ITH) as a major underlying cause of therapy resistance and cancer recurrence[1, 2]. At the genetic level, ITH is most frequently measured by 'bulk' sequencing of tumor biopsies[3], or by single cell sequencing (a limited number of) individual cells sorted by flow cytometry[4]. However, these approaches may fail to identify rare, potentially clinically relevant subclones if they are present at a frequency below the limit of next generation sequencing (NGS) detection or if they are incorrectly classified as artefacts. A further limitation of these approaches is that they do not preserve the tissue architecture; therefore the spatial and histopathological context of tumor subclones is lost. This information contains crucial clues as to the mechanisms of subclone emergence, and thus can provide valuable and otherwise inaccessible insight into tumor evolution.

In situ mutation detection (ISMD) is therefore a useful tool for the study of tumor evolution and heterogeneity that preserves spatial context. It may also be utilized to complement NGS data, for example to confirm the presence of low frequency variants, or in the case of mRNA-ISMD to confirm that a mutant transcript is indeed expressed. Furthermore it could be applied diagnostically as a tool for cost-effective and high-throughput screening of

samples of unknown mutational status, perhaps with the goal of directing therapeutics. Existing ISMD techniques are, however, limited by their technical complexity, and their lack of sensitivity and/or specificity (see Discussion), and therefore have not been widely adopted by researchers or in clinical practice. Here we describe in situ analysis of mutational status at a cellular level using novel BaseScope technology. The BaseScope assay is an RNA in situ hybridization (ISH) technique, based on the same principles as the well-established RNAscope mRNA expression assay[5], involving a unique probe design and signal amplification strategy. The technology utilizes 'Z' probes, each of which contains an 18- to 25-base region complementary to the RNA sequence of interest, followed by a spacer sequence, then a short "tail" sequence that is recognized by the signal amplification system. The key concept in this technology is that signal amplification requires two 'Z' probes binding adjacent to each other in order for a signal-generating "tree" to form at the target site. We have shown this approach to be a highly sensitive and specific approach to analysis of mRNA expression[6, 7]. Compared to RNAscope, BaseScope incorporates an additional signal amplification step to further boost detection sensitivity without increasing background noise. As a result, BaseScope requires only

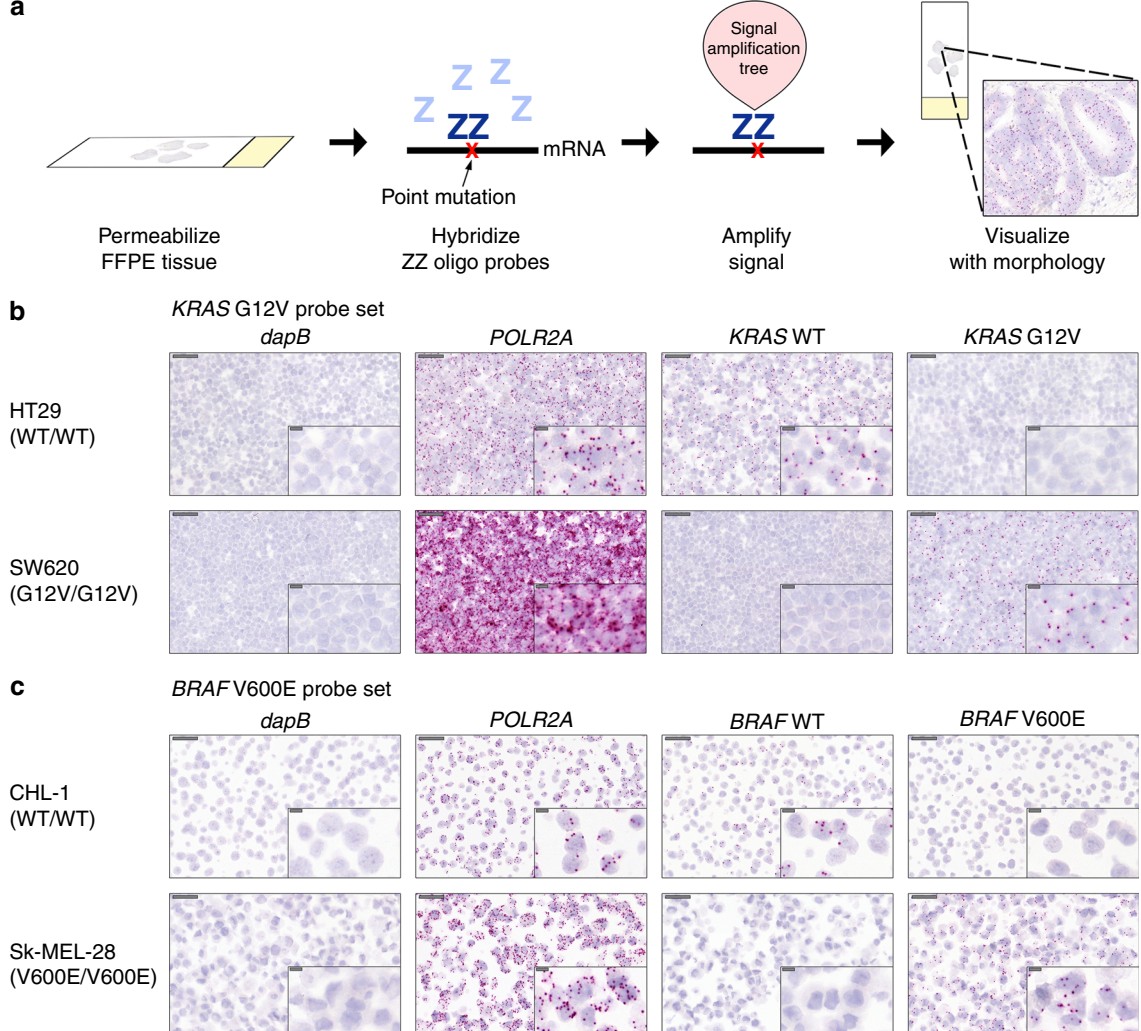

**Fig. 1** Validation of BaseScope probes in cell lines. **a** Schematic of BaseScope technology. Binding of two custom-designed 'Z' probes to target mRNA directs binding of a signal amplification tree, allowing visualization of point mutations by FastRed. **b, c** Representative images of the validation of the *KRAS* G12V probeset **b** and *BRAF* V600E probeset **c** In both cases a wild-type cell line and a homozygous mutant cell line are shown, with a negative control probe (*dapB*), a positive control probe (*POLR2A*), the wild-type probe and the mutant probe. Probe binding is visualized as punctate red dots. Scale bars represent 50 micron and 10 micron (inset)

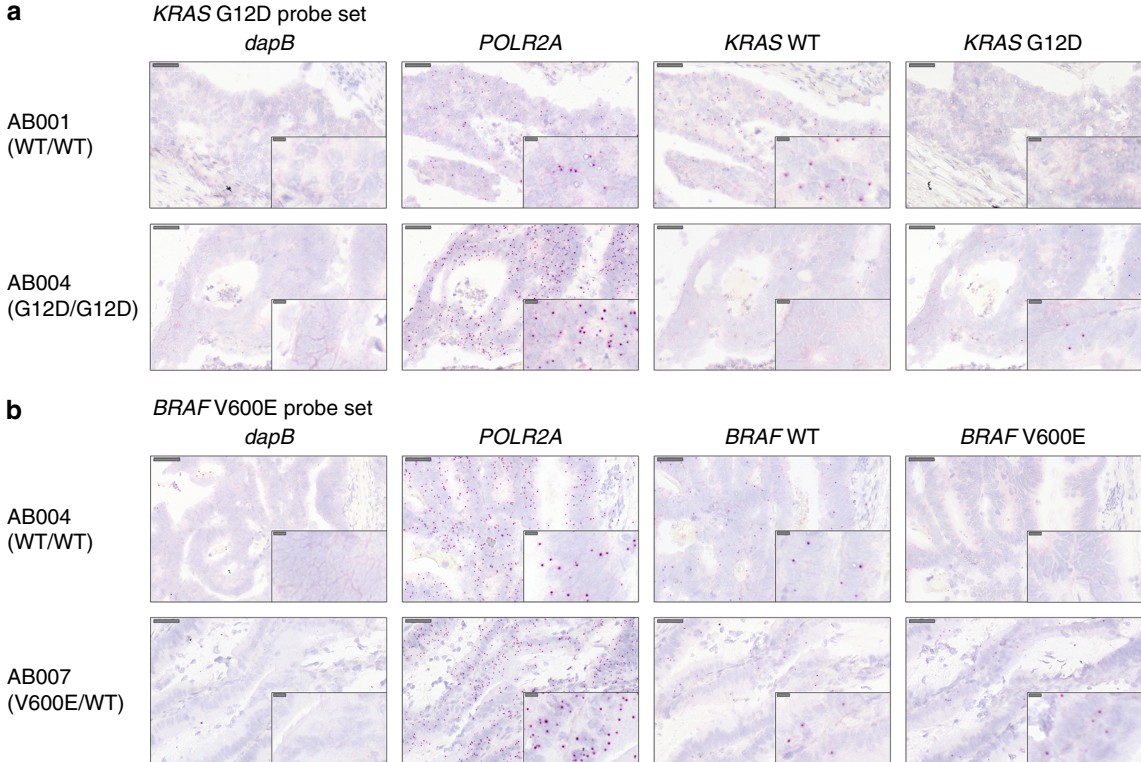

**Fig. 2** Validation of BaseScope probes in tumors. Representative images of the validation of the *KRAS* G12D probeset **a** and *BRAF* V600E probeset **b** in archival human tumor tissue. In both cases a wild-type tumor and a mutant tumor are shown, with a negative control probe (*dapB*), a positive control probe (*POLR2A*), the wild-type probe and the mutant probe. Scale bars represent 50 micron and 10 micron (inset)

a single ZZ pair ('1ZZ') instead of the standard 20 ZZ pairs used in RNAscope, allowing detection of short RNA sequences and the discrimination between single nucleotide alterations (Fig. 1a). Here, we demonstrate the sensitivity and specificity of custom-designed, mutation-specific 1ZZ BaseScope probes for RNA-based ISMD in cell lines and primary tumors.

We study the expansion of mutant subclones in colorectal cancer (CRC). CRCs have a strong stereotypical histological structure—they consist of thousands of individual glands, each containing a few thousand cells that are thought to be recently clonally derived from a smaller number of long-lived lineages within the gland[8–11]. Previously, we have identified spatially-contiguous subclonal expansions of glands within colorectal adenomas using putatively neutral mitochondrial enzyme defects to trace clones[8, 12]. The inter-gland clonal architecture of CRCs is less well resolved: gland-by-gland measurement of methylation patterns provided gross-scale spatial maps of clonal ancestry, which mathematical modeling predicted were consistent with a rapid initial burst of tumor growth followed by relative quiescence[13]. Subsequently, genome-wide copy-number analysis on a gland-by-gland level revealed occasional extensive clonal mixing within CRCs[9], and a targeted DNA copy number analysis suggested a complex 3D tumor architecture in a single extensively mapped case[14]. Because these previous genotyping methods have required laborious tissue microdissection, detailed topographical maps of clone spread throughout CRCs are lacking.

Here we validate BaseScope ISMD technology and use it to map the topography of subclones bearing driver mutations in one of the *KRAS*, *PIK3CA* or *BRAF* proto-oncogenes in CRCs and adenomas.

## Results

**Validation of 1ZZ BaseScope probes in human cell lines**. We characterized a total of 9 probe sets designed to target driver gene

point mutations commonly found across human cancers: *KRAS* (G12D, G12V, G12A, G12S, G12R, and G12C), *BRAF* (V600E) and *PIK3CA* (E545K and H1047R, for probe details see Supplementary Table 1). Validation was performed in well-characterized human cell lines known to be mutant or wild-type for the mutation of interest (Supplementary Table 2). All assays included probes targeting the mutant sequence and the corresponding wild-type sequence, together with positive and negative RNA quality controls (negative control: bacterial mRNA *dapB*, positive control: housekeeping gene *POLR2A*). BaseScope signals were readily distinguishable as punctate red dots within cells (Fig. 1). For each probe in each cell line, the proportion of cells containing positive signal was quantified, with an average of 383 ($\pm$27) cells counted per slide.

Firstly we validated the sensitivity and specificity of the 1ZZ BaseScope technology by quantifying the staining of the positive and negative control probes (Fig. 1, Supplementary Fig. 1 and Supplementary Table 2). Across the cell lines examined ($n =$ 16 slides representing 11 cell lines), an average of 99.1% ($\pm$0.4%) of cells showed signal for the positive control probe (*POLR2A*) whereas an average of 0.29% ($\pm$ 0.20%) of cells displayed signal for the negative control probe (*dapB*) indicating high sensitivity and specificity, respectively.

To test the sensitivity of the mutation-specific 1ZZ BaseScope probes, we applied them to the cell lines of known mutational status (Fig. 1, Supplementary Fig. 1 and Supplementary Table 2). In homozygous mutant cell lines ($n = 4$), an average of 85.6% ($\pm$ 11.6) of cells were positive for expression of the mutation, and in heterozygous mutant cell lines ($n = 5$), an average of 50.5% ($\pm$ 16.9) of cells were positive. We note that there is likely natural (biological) variation in gene expression meaning that not all cells necessarily express the mutant transcript at detectable levels, despite carrying the mutant gene. Technical repeats on three cell

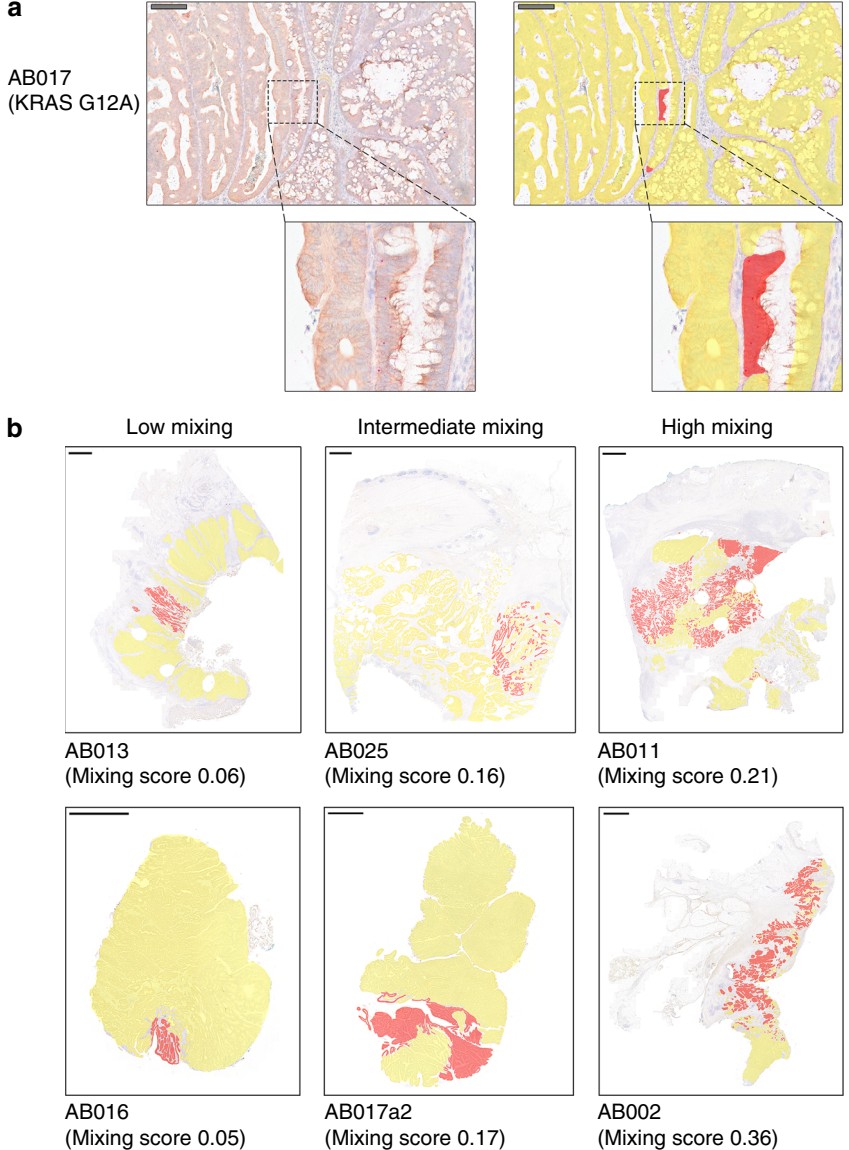

**Fig. 3** BaseScope detection of CRC subclonal architecture. **a** Representative images showing BaseScope detection of a small patch of mutant cells (red) within a field of wild-type cells (yellow). Scale bars represent 200 micron. **b** Representative topographical maps of subclonal architecture showing the spatial arrangement of mutant (red) and wild-type (yellow) subclones. Examples of low mixing (left column), intermediate mixing (middle column) and high mixing (right column) are shown. Scale bars represent 2000 micron

lines, carried out at least 3 months apart, showed that this signal was highly reproducible (no significant difference in signal quantification between experimental repeats by the Mann-Whitney test; $n = 3$ cell lines, Supplementary Fig. 2a). We then tested the specificity of the mutation-specific BaseScope probes by looking for positive signal in cell lines known to be wild-type for the probe-specific mutation. Across the 9 probe sets, a non-specific (putatively false positive) signal was observed in 0.08% ($\pm 0.08$%) of cells. Finally, the homozygous mutant cell lines ($n = 4$) were used to determine the specificity of the wild-type BaseScope probes, and an average non-specific 'false positive' signal was observed in 0.19% ($\pm 0.27$%) of cells.

**Validation of 1ZZ BaseScope probes in human tumors**. We next evaluated the efficacy of the BaseScope technology in archival human tumors of known mutation status (Supplementary Tables 3 and 4). The negative control probe performed similarly to the cell lines: non-specific signal (*dapB*) was detected

in 0.18% ($\pm 0.21$%) of cells ($n = 5$ representative tumors). However the fraction of cells positive for the positive control (*POLR2A*) was variable between tumors, with an average of 66.0% ($\pm 24.8$%) of cells ($n = 4$ representative tumors) displaying positive signal (Fig. 2 and Supplementary Table 4). This is likely because of varying degrees of mRNA degradation that may occur during processing and storage of FFPE material, together with potential inter-tumor heterogeneity in *POLR2A* mRNA expression.

Staining for mutation-specific BaseScope probes was entirely consistent with prior knowledge of the mutational status of the tumors (Fig. 2 and Supplementary Table 3). The proportion of mutation-positive tumor cells in mutant samples was variable with an average of 24.5% ($\pm 12.5$%; $n = 4$ tumors) whereas reassuringly the proportion of mutant signal detected in wild-type samples was very low at 0.15% ($\pm 0.29$%, $n = 3$ tumors). The positive cell fraction was highly reproducible between technical repeats (Supplementary Fig 2a). The tumor-stroma interface

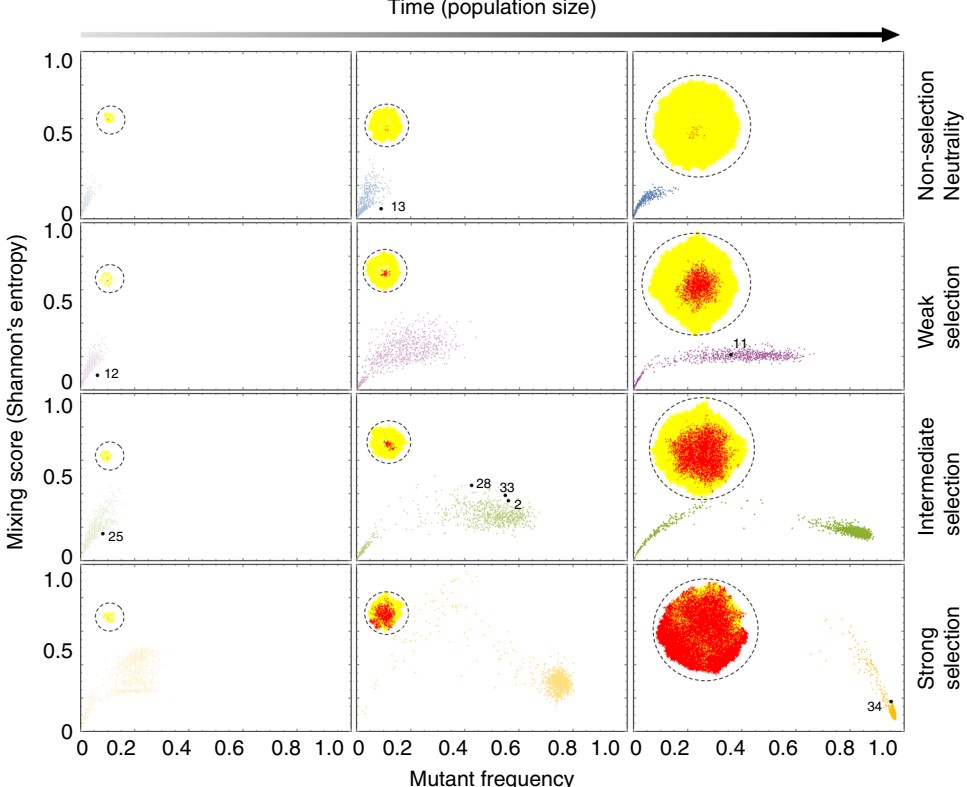

**Fig. 4** Inference of subclone evolutionary dynamics. Computational modeling predicts the relationship between clone intermixing and clone size, and the time of clone appearance and selective advantage. Clone mixing level increases to a transient high at intermediate times as the subclone expands, and decreases as the clone takes over the tumor or is lost (time increases along the columns). The strength of selection experienced by the subclone determines the speed and maximal extent of subclone mixing (selection increases down the rows). Black dots are empirically measured mixing values in primary tumors, and are plotted for the parameter values that explain the data with maximum posterior probability (see Methods). The inner panels (dashed circles) are representative simulation snapshots indicating the expected clonal intermixing. Each panel shows results from 1000 independent realizations; relative growth ratios of the mutant to wild type are 1, 1.75, 3.0, 8.0 for neutral, weak, intermediate, and strong selection, respectively; the initial mutant frequency is 0.01 and the final tumor size is $10^4$

provided another demonstration of probe specificity: for example, in a *KRAS* G12D homozygous mutant tumor, positive signal from the *KRAS* wild-type probe was only evident in stromal cells (Supplementary Fig. 2b).

We next sought to determine if BaseScope probes were also effective in mutational screening of samples of uncertain mutational status by using a pool of probes against the *KRAS* mutations G12D, G12V, G12A, G12C, and G12R. We performed pooled BaseScope staining on 21 colorectal adenomas containing foci of cancer ('Ca-in-ad' tumors, see Methods and Supplementary Table 3) which all appeared *KRAS* wild-type by Sanger sequencing from 'bulk' DNA. The pooled probes detected *KRAS* mutations at subclonal frequency (for details see subsequent Results sections) in 3/18 samples, and we then assayed these samples with the individual probes to determine which specific mutation was present. Importantly, we found the probes to be highly specific to the individual mutation, displaying no cross-reactivity with other mutant probes (Supplementary Fig. 2c). We subsequently validated the de novo mutations in all three cases using microdissection (Supplementary Fig. 2d) of mutant and wild-type regions and Sanger sequencing.

Finally, we combined BaseScope with a concurrent in situ analysis of protein or DNA. Sequential immunohistochemistry (IHC) for cytokeratin expression and DNA fluorescence in situ hybridization (FISH) for Chr18q were successful and had no effect on the tissue morphology or the quality of the BaseScope signal (Supplementary Fig. 3).

**Use of BaseScope 1ZZ probes to detect subclonal architecture**. The spatial distribution of subclonal driver mutations provides insight into the mechanisms of tumor growth and evolution[9, 15]. We sourced archival CRC tumors ($n = 8$) that had been shown by the assessment of variant allele frequency (VAF) derived from NGS to contain possible subclonal driver mutations in *KRAS*, *BRAF*, or *PIK3CA*. Further samples ($n = 3$ 'Ca-in-ad' tumors) had been shown to carry subclonal *KRAS* mutations using the aforementioned pool of *KRAS* mutant probes. BaseScope technology was used to identify subclones, and digital 'subclonal maps' depicting the spatial topology of subclones were generated from images of 13 lesions from 11 patients (Fig. 3; Supplementary Fig. 4). Subclones differed greatly in size between tumors (range 3–95% of total tumor), and tiny clonal patches of 10 or fewer cells were found within fields of otherwise wild-type cells (Fig. 3a). The analysis of serial sections permitted 3D reconstruction of clone shape, which revealed intimate mixing of subclones (Supplementary Movie 1 and Supplementary Fig. 5).

In each of the three 'Ca-in-ad' cases, the mutant subclone was localized to a histologically distinct region. In one case (AB019) the mutant appeared clonal within the invasive region, with the non-invasive adenomatous region exclusively *KRAS* wild-type. In the two remaining cases (AB016 and AB017), the *KRAS* mutant population was at a subclonal frequency within the adenomatous region, with the carcinoma exclusively *KRAS* wild-type. The finding of *KRAS*-mutant clones exclusively within precursor adenomatous regions of a cancerous lesion suggests that *KRAS*

mutation status alone could be insufficient to select patients for EGFR inhibition, and instead the histological location of the mutation may provide additional predictive information about drug response.

To characterize the phenotype of the morphologically indistinguishable subclones ('CRCs', $n = 8$, see Supplementary Table 5), we examined wild-type and mutant regions for expression of Ki67 (proliferation), and CD8 (cytotoxic T lymphocyte infiltration, Supplementary Fig. 6a). We detected cases in which the mutant subclone displayed distinct molecular phenotypes: in case AB034, for example, the *BRAF* V600E mutant subclone displayed significantly elevated proliferation ($P = 0.008$ by the Mann–Whitney test, Supplementary Fig. 6b), and in cases AB002 and AB025, the *KRAS* G12V mutant subclones displayed significant increases in infiltrating cytotoxic T cells ($P = 0.02$ and $P = 0.03$ respectively by the Mann–Whitney test, Supplementary Fig. 6b). There was a weak association between Ki67 positivity and cytotoxic T cell infiltration ($R^2 = 0.29$, $P = 0.033$ by the F test, Supplementary Fig. 6c).

We noted highly heterogeneous patterns of subclonal architecture, with some cases displaying clear clonal boundaries and others displaying a high level of mixing with wild-type cells (Fig. 3b). The degree of mixing was quantified using Shannon's entropy (Supplementary Fig. 7a, Supplementary Table 5). Three samples (AB002, AB028, and AB033) showed particularly high mixing scores (Supplementary Fig. 7b). The proportion of tumor occupied by the subclone and the mixing score were not correlated ($P > 0.05$ by the F test, Supplementary Fig. 7c).

**Subclones arise early or have a large fitness advantage**. To investigate the mechanisms underlying the observed clonal architectures we computationally modeled the spread of mutant subclones during tumor growth. Clones expanded by dividing into vacant space, or pushing neighboring cells to create space. We simulated scenarios where a driver mutation arose early or late in tumorigenesis, with various selective advantages compared to the wild-type. We monitored the mutant frequency and the degree of clone intermixing (measured by Shannon's entropy) over time. As our experimental observations suggested, the computational model confirmed that mutant frequency and mixing score have a complex, non-linear relationship (Supplementary Fig. 8a).

In the simulations, neutral mutants rarely underwent large clonal expansion (Fig. 4, upper panels) whereas strongly selected mutants could fix (take over) in the tumor (Fig. 4, lower panels). Irrespective of selection, the level of mixing was initially low when the new clone was formed, increased to a transient high, and finally decreased to a low value as the mutant clone either took over the population or was lost. The strength of selection determined the speed of this transient behavior, and the maximal extent of mixing observed (Fig. 4; Supplementary Fig. 8). The level of intermixing observed in most primary tumors was broadly consistent with weak or intermediate selection, or sampling the tumor at an intermediate time relative to mutation introduction under strong selection (Fig. 4; Supplementary Fig. 9). A case with a low mixing score (e.g., AB016) was most likely to be explained by an early arising mutant with an intermediate fitness advantage, and conversely a sample with a higher mixing score (e.g., AB017a1) is more likely to have had a later arising mutant and with a larger fitness advantage (Supplementary Fig. 8b–e). We confirmed our findings in an alternative model, whereby cells divided without pushing and only filled neighboring empty spaces vacated by cell death. The temporal patterns of clone spread and mixing were qualitatively the same in both formulations of the model (Supplementary Fig. 10).

## Discussion

This manuscript describes the first application of the novel 1ZZ BaseScope in situ hybridization technology to measure the spatial distribution of point mutations in human archival FFPE samples. Our analysis shows the very high sensitivity and specificity of 9 BaseScope probesets, representing common driver mutations in the genes *KRAS* (6 probesets), *BRAF* (1 probeset) and *PIK3CA* (2 probesets). The sensitivity and specificity of BaseScope probes allowed for precise spatial mapping of mutant subclones in human tumors. These spatial maps reveal the heterogeneity of clonal mixing patterns in CRC, and we report examples of highly infiltrated, mixed clones, as well as clones with distinct boundaries. Computational modeling shows how these spatial data are indicative of the time of subclone appearance and the relative fitness of the subclone.

We note that BaseScope detected minor subclones with an estimated frequency as low as 3%, and these would likely not be reliably detected by standard 'bulk' sequencing. In fact, we report a number of samples that were previously shown to be *KRAS* wild-type by Sanger sequencing, but were shown to contain validated *KRAS* mutant subclones by BaseScope. This high sensitivity of subclone detection could have important implications for the stratification of patients for targeted therapy, for example for EGFR blockade in CRC that requires the tumor to be KRAS wild-type to be efficacious[16]. Further work is required to investigate the clinical utility of the method.

Our results suggest that BaseScope provides a robust technology for ISMD, a technique that traditionally has been performed using DNA or RNA in situ PCR with mutation-specific primers[17]. Although they can be effective, in situ PCR approaches such as rolling circle amplification of padlock probes[18] are often time consuming, as they first require conversion of mRNA to DNA by synthesis of a complementary nucleic acid strand prior to hybridization. Furthermore they are technically challenging, as each probe requires different PCR and hybridization conditions and extensive optimization. Using BaseScope technology, ISMD can be performed in approximately 8 h, hybridization conditions are universal, and minimum assay optimization is required by the user. Additionally, as BaseScope generates a chromogenic end product instead of a fluorescent end product, there are no specialist microscopes required for analysis, and stained slides can be stored long term. Moreover, we have shown that BaseScope can be readily combined with sequential IHC or DNA FISH, in order to concurrently study protein-level or DNA-level alterations in situ. Recent technology that allows for combining mutation detection with copy number analysis includes specific-to-allele PCR–FISH (STAR-FISH[19]). In our hands we have found concurrent ISMD and copy number detection by BaseScope and DNA-FISH offers improved sensitivity, reliability and reproducibility over STAR-FISH, as well as a reduced assay time of 24 h (vs. ~40 h).

The BaseScope methodology has a small number of potential limitations compared with other methods. First, the reliance on mRNA means that the mutant mRNA must be expressed at a detectable level in the tissue or cell of interest, and so mutations within untranscribed regulatory sequences such as promoters will not be detected by RNA-ISH with BaseScope. Second, some archival cancer samples may fail owing to poor RNA quality, although this problem is likely to be restricted to older samples that have undergone sub-optimal processing. Reassuringly all 51 archival samples (average time since fixation = 6.4 years, range 0–20 years) analyzed in this study were of sufficient quality for analysis by BaseScope. Third, whilst the method has high specificity, its sensitivity for detecting heterozygous mutations is typically ~25%. In practice, this has minimal effects on the detection of even very rare subclones, but the method does require some development to be suitable for true single cell analysis. We note that the BaseScope

signal is likely to be proportional to the underlying copy-number of the locus. Finally, some protein-inactivating mutations cause unstable mRNA–while the specific mutations will not be detected in this scenario, we would nonetheless detect reduced or absent transcript levels by BaseScope using the wild-type control probe. These limitations can potentially be addressed by a DNA version of the BaseScope assay in the future.

This manuscript is the first to describe the application of point mutation specific 1ZZ BaseScope in situ hybridization technology. The method provides a robust and efficient approach to in situ mutation detection. It has broad application in cancer biology research and the potential for clinical application, including sensitive identification of resistant clones in cancers, mapping of mutations to benign and malignant components of mixed tumors and molecular cancer staging.

## Methods

**Cell lines**. Cell lines were obtained from the American Type Culture Collection (ATCC; www.atcc.org). Frozen aliquots were supplied to AbboMax Inc. (San Jose, CA) where they were processed into highly consistent and homogenous Cellmax pellet blocks. A list of cell lines used can be found in Supplementary Table 2.

**Patient samples**. Formalin-fixed paraffin embedded (FFPE) tissue blocks (n = 51, Supplementary Table 3) were obtained from University College and St Mark's Hospitals, London, under multi-center ethical approval (London Stanmore committee, 11/LO/1613), with all patients giving informed consent. Additional samples were from the Manchester Cancer Research Centre Biobank (Project 13_NIWR_01); the QUASAR2 trial (ISRCTN45133151, ethical approval REC 09/H0606/5+5); the COIN trial (ISRCTN; 27286448); and the FOCUS trial (ISRCTN; 79877428, ethical approval 15/EE/0241). The average time since fixation of the tissue blocks was 6.4 years (range 0–20 years). Samples were examined by at least two expert histopathologists.

**BaseScope assay**. BaseScope assays were performed in accordance with guidelines provided by the supplier (Advanced Cell Diagnostics, Newark, CA). Sections were taken at 5 μm thickness onto Superfrost plus slides (Fisher Scientific, Loughborough, UK) and allowed to dry overnight at room temperature (RT). Sections were then baked at 60 °C for 1 h before deparaffinizing in xylene (2 × 5 min) and ethanol (2 × 2 min), then drying by baking at 60 °C for 2 min. Pretreat 1 (hydrogen peroxide) was applied for 10 min at RT, Pretreat 2 (target retrieval) for 15 min at 100 °C and Pretreat 3 (protease) for 30 min (tissue sections) or 15 min (cell pellets) at 40 °C, with two rinses in distilled water between pretreatments. BaseScope probes (see Supplementary Table 1) were then applied for 2 h at 40 °C in a HybEZ oven before incubation with reagents AMP0 (30 min at 40 °C), AMP1 (15 min at 40 °C), AMP2 (30 min at 40 °C), AMP3 (30 min at 40 °C), AMP4 (15 min at 40 °C), AMP5 (30 min at RT) and AMP6 (15 min at RT). Slides were rinsed with wash buffer (2 × 2 min) between each AMP incubation. Finally slides were incubated with Fast Red for 10 min at room temperature in the dark. Slides to be used for sequential staining were kept in PBS at 4 °C overnight before proceeding to the next protocol. Slides that were not used for further staining were counterstained with Gill's hematoxylin before drying for 15 min at 60 °C and mounting in VectaMount permanent mounting medium (Vector labs, Burlingame, CA).

**Immunohistochemistry (IHC)**. After performing the BaseScope protocol, slides were blocked in 2% goat serum with 1% bovine serum albumin (BSA) for 1 h at RT. Primary antibody anti-human cytokeratin (#M3515, Agilent Technologies, Santa Clara, CA) was applied for 1 h at RT (1:2000 dilution), followed by biotinylated rabbit anti-mouse secondary antibody (#E0354, Agilent Technologies) for 30 min at RT (1:400 dilution) then streptavidin-conjugated horseradish peroxidase (#P0397, Agilent Technologies) for 30 min at RT (1:500 dilution). Slides were developed in 3,3′-diaminobenzidine (DAB) for 2 min, and counterstained with Gill's hematoxylin before drying for 15 min at 60 °C and mounting.

For dual-color IHC, 5 μm serial sections were dewaxed, rehydrated and immersed in 3% hydrogen peroxide for 20 min to quench endogenous peroxidase activity. Antigen retrieval was carried out at 95 °C for 20 min in sodium citrate buffer (pH 6.0). After cooling, sections were incubated with blocking buffer (PBS supplemented with 2% goat serum and 1% BSA) for 1 h at RT. Primary antibodies (Ki67, #ab92742, Abcam at 1:2000 dilution) were diluted in blocking buffer and applied for 1 h at RT. Sections were then incubated with a biotinylated secondary antibody at RT for 45 min, followed by incubation with streptavidin-biotin peroxidase solution at RT for 45 min. Visualization of the first antibody binding was carried out using DAB. Slides then underwent a second round of antigen retrieval, generally at 95 °C for 5 min in sodium citrate buffer (pH 6.0), before applying the blocking buffer for a further 1 h at RT. The second primary antibody was then applied (CD8, #7103, Dako at 1:100 dilution), followed by incubation with a biotinylated secondary antibody at RT for 45 min and incubation with

streptavidin-alkaline phosphatase. Visualization of the second antibody binding was performed using Fast Red, according to the manufacturer's instructions (Abcam, Cambridge, UK). Finally, sections were lightly counterstained using Gill's hematoxylin, and allowed to dry before mounting.

For manual scoring of IHC, three representative regions of 100 tumor cells were considered for the wild-type and mutant regions. For Ki67 staining, the proportion of nuclei displaying strong positivity was quantified. For CD8 staining, the number of positively stained cells infiltrating into, or in close proximity to the tumor was quantified.

**Fluorescent in situ hybridization**. After performing the BaseScope protocol, Fluorescent in situ hybridization (FISH) was performed on the same tissue section using the FISH accessory kit (Agilent Technologies) according to the manufacturer's guidelines. Sections were incubated overnight at 37 °C with a SureFISH probe against Chr18q21.2 (#G100219G-8, Agilent Technologies). Nuclei were counterstained with DAPI, before visualization on the Ariol system (Leica Biosystems, Milton Keynes, UK). The BaseScope Fast Red signal is fluorescent in the red channel, and the FISH probe in the green channel.

**Mutation validation by sequencing**. Sections of 12 μm thickness were taken serial to the sections used for the BaseScope assay. For needle macrodissection the sections were placed onto glass slides, and for laser capture microdissection (LCMD) onto UV-treated PEN-membrane laser capture slides (Carl Zeiss Microscopy, Göttingen, Germany). Slides were deparaffinized, rehydrated and lightly stained with methyl green before scraping or LCMD of mutant and wild-type regions of tumor. DNA was extracted using the QiaAMP DNA micro kit (QIAGEN Ltd, Manchester, UK) and used for a two-round nested polymerase chain reaction (PCR) to amplify KRAS exon 1. PCR conditions and primer sequences were. as previously described[20] (First round: forward primer GAGTTTGTATTAAAAGG-TACTGGTGGA, reverse primer ATCAAAGAATGGTCCTGCAC, 35 cycles 95 °C, 30 s, 60 °C 30 s, 72 °C 30 s. Second round: forward primer TTTGA-TAGTGTATTAACCTTAT, reverse primer TATTAAAACAAGATTTACCTC, cycles: 95 °C 30 s, 55 °C 30 s, 72 °C 30 s). Sequencing was performed using BigDye v3.1 terminator cycle sequencing chemistry on the ABI 3730 DNA analyzer (Applied Biosystems Inc., Foster City, CA).

**2D/3D mapping of subclonal mutations**. For 2D mapping, sections were stained with BaseScope using the relevant 1ZZ mutant probe, and then IHC for cytokeratin was performed, as described above. Slides were digitized using the Pannoramic 250 scanner (3D Histech, Budapest, Hungary). Wild-type and mutant glands were manually annotated on each section using Adobe Photoshop CS6, under the assumption that mutant glands are clonal for the mutation of interest. The subclone proportion was calculated by dividing the pixel count in mutant regions by the total pixel count in mutant and wild-type regions. Subclones were validated by an experienced pathologist as being morphologically distinct ('Ca-in-ads', n = 5) or morphologically indistinguishable ('CRCs', n = 8). For the 3D map, 120 sections were taken at 7 μm thickness, then every tenth section was used for the BaseScope assay with the probe against the KRAS G12A mutation. Scanned data sets were uploaded and registered in FreeD[21] for 3-dimensional reconstruction, as follows. Serial images (TIFF file format) were imported into FreeD software v 1.10 image stack files. Gland boundaries were drawn manually in each 2D serial image and connected along the third dimension between adjacent slides. This procedure is facilitated by simultaneous display of masked gland boundaries during virtual microscopy in FreeD software. After manual 2D assessment of all virtual tissue slides of one stack, 3D models can be computed and visualized by interconnection of the defined masks along the third dimension in FreeD software.

**Calculation of Shannon's entropy**. The Shannon entropy metric was used to quantify spatial heterogeneity of mutant and wild-type populations. Mathematically the Shannon entropy is defined as $S = -\sum_{i=1}^{n} \left[ p_i \log_2 p_i + (1 - p_i) \log_2 (1 - p_i) \right] / n$, where the tumor image has been divided into $n$ quadrats of the same size and $p_i$ is the frequency of the mutant type in quadrat $i$. If the mutant and the wild type are fully mixed, the Shannon's entropy of the tumor population is 1, whereas if the mutant and wild type cells are completely spatially separate the Shannon's entropy is 0.

For 2D maps of human CRCs, annotated digitized sections were overlaid with a grid comprising quadrats of 1300 μm² (average 96 quadrats per case, range 16–216). Within each quadrat, the percentage of pixels that were yellow (wild-type) or red (mutant) was calculated using ImageJ[22]. These measurements were then used to calculate Shannon's entropy for each lesion, as described for computer-generated tumors.

**Computational model of subclonal tumor growth**. We explored the effects of two important evolutionary forces on the spatial patterns of mutant and 'wild-type' subclones, namely the strength of subclonal selection and the time when the mutant arises. We modeled the growth of tumor populations and the appearance of mutants using a spatial stochastic branching process. Cells occupied a simple lattice with eight nearest neighbors. When a tumor cell divided, one daughter cell was placed at random in an empty neighboring position and the other daughter cell remained in the position of the parent cell. If there was no empty position around

the dividing cell, a random direction was chosen and cells were pushed by one position in that direction to create space for the new daughter cell. We also considered an alternative model whereby cells could only divide into a empty neighboring space (i.e. no pushing), and where all cells die randomly at a fixed rate δ. Neutral evolution was defined as the scenario where the mutant and wild-type cells had the same division rate, and selection quantified by the relative increase in the rate of at which the mutant subclone attempted to divide. We also considered various arising times (early or late) for the mutant population.

**Statistical inference of subclonal dynamics**. We employed an Approximate Bayesian Computation (ABC) approach[23]. For each CRC we simulated the computational model many times. Each simulation took input parameters that described the time of subclone arrival and its relative fitness: these parameters selected from uniform prior distributions, as described in Supplementary Fig. 9. For each simulation, we calculated how well the simulated data matched the empirical data, using the Euclidean distance between the observed vs. computed mutant frequency and mixing score. Simulation instances that produced a Euclidean distance $\varepsilon < 0.05$ were 'accepted' into the posterior distribution. The parameter values with maximal posterior probabilities were reported for each CRC.

**Statistics**. Results in the text are reported as mean ± 95% confidence intervals unless stated otherwise. Parametric and non-parametric tests were used as appropriate, defined in the figure legends.

**Data availability**. Data generated and analyzed in this study are available from the corresponding authors upon reasonable request.

**Code availability**. Computational code used in this study is available from the corresponding authors upon reasonable request.

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

## Acknowledgements

We would like to thank George Elia and Emily Austen (BCI histopathology) for expert sectioning of FFPE material. This work was supported by Cancer Research UK (A14895, A.-M.B. and N.A.W.; A21870 A.L. and I.T.; A23923, A.D.B.; A22745, M.J.; A19771, T.A. G.) and the NIHR University College London Hospitals Biomedical Research Centre (MR-J). We also acknowledge the EU ERC EVOCAN and INSITUMUT projects (I.T.), the Wellcome Trust (102732/Z/13/Z, ADB; 202778/Z/16/Z, TAG; 202778/B/16/Z, A.S.) and core grant 090532/Z/09/Z to the Wellcome Trust Centre for Human Genetics from the Wellcome Trust (A.L., I.T.). A.S. is supported by the Chris Rokos Fellowship in Evolution and Cancer and by Cancer Research UK (C52506/A22909). A.S. and T.H. are supported by the Wellcome Trust grant to the Centre for Evolution and Cancer (105104/ Z/14/Z). For tissue provision we acknowledge the UCL/UCLH Biobank for Health and Disease the FOCUS trial (Chief investigator Matt Seymour, University of Leeds and the MRC Clinical Trials Unit at UCL) and the S:CORT consortium (Phil Quirke and Susan Richman, sample curation and histopathology; Ultan MacDermott, Aikaterini Chatzipli and the Wellcome Trust Sanger institute for genetic sequencing on FOCUS trial samples within S:CORT; IT, ED, T.M. and A.B. for data analysis within S:CORT). S:CORT is an MRC and Cancer Research UK stratified medicine consortium for colorectal cancer (grant award no MR/M016587/1).

## Author contributions

A.-M.B. performed the experiments. X.-M.M.W., X.-J.M., J.K., C.M.A., X.W., L.P., N.S., Y.L., A.L. and E.P. were responsible for developing and optimizing the BaseScope technology, and providing human cell line pellets. W.H. developed the computational models of tumor growth, and calculated mixing scores. M.J. produced the 3D map of subclonal interactions. T.H. and A.S. provided bioinformatics support. M.J., N.A.W. and M.R.-J. provided expert histopathological guidance. E.D.,A.D.B., M.J., M.R.-J. and I.T. identified and provided human tissue samples. A.-M.B., E.P., I.T. and T.A.G. were responsible for study design and implementation. A.-M.B., W.H., I.T. and T.A.G. wrote the manuscript. All authors read and approved the manuscript.

## Additional information

**Competing interests**: X.-M.M.W., X.-J.M., J.K., C.M.A., X.W., L.P., N.S., Y.L. and E.P. are employees of Advanced Cell Diagnostics, Inc. The remaining authors declare no competing financial interests.

**Publisher's note**: 

