## [Peer Review File · Nature Communications]

Reviewers' comments:

Reviewer #1 (Remarks to the Author):

In the present manuscript Baker et al. introduce an improved technique of RNA in situ hybridization. The validation of the technology seems robust. Using this technique the authors continue to identify sub-clonal mutations, which potentially could be missed by bulk sequencing with limited depth. The technique can be an important tool to study intra-tumor heterogeneity, a possible cause of resistance to drug therapy. In short, the method is well demonstrated. Unfortunately the usefulness of its application beyond detection of small clones is not clear.

To facilitate the implication of the technique by others, it would be helpful if the authors explain in some more detail how the technique differs from BaseScope and how the reduction to single ZZ base pairs is achieved.

A limitation of the introduced technique is that for each specific (point) mutation, a specific probe is required. It therefore appears that NG bulk sequencing to identify mutations needs to precede RNA ISH, or ISH needs to be performed with a large set of point mutation specific RNA probes. How do the authors envision that this technique can best be applied on naive tumors? In addition, how does this technique deal with sampling errors as only a very limited proportion of the tumor can be analyzed?

One major challenge in studying intra-tumor heterogeneity is identifying the relative fitness and emergence of a clone from resected material. The problem is that from resected data only the final clone size can be observed, while the fitness advantage, the time at which the clone emerged and random fluctuations all contribute to the final clone size. The authors aim to address this problem, as well as clonal mixing, by using a computational model. I have several more detailed questions regarding this part of the manuscript:

- 1) Do the authors suggest that they can distinguish between the timing and the relative fitness of clones? While this could potentially be very interesting, the manuscript is not clear on this point. As a key observable the authors use clonal mixing. Can they explain how mutational fitness and timing of a clone differ in the pattern of clonal mixing that arises? How sensitive are the results to the details of the model?
- 2) Why is clonal mixing being used? What is the advantage over the mutation frequency? What is the relation between these two observables in the model? And how would changes in adhesion or migration (which are not captured in the model) affect the clonal mixing, given a certain mutation frequency?
- 3) The dynamics of cells in their simulation seems unrealistic. Adenocarcinoma's are thought to at least partially grow by a process of gland fission, not only random movement of cells. The assumed dynamics probably strongly influences the clonal mixing. What is the motivation for the chosen model/dynamics?
- 4) In Fig. 4 the results are plotted against time. Is it the relative time at which a clone occurs or the absolute time, i.e. the age of a tumor that is studied here? If the latter is the case, can the authors translate the inferred value to an actual age of the tumor?

Reviewer #2 (Remarks to the Author):

The paper by Baker et al. describes the application of the BaseScope assay for in-situ mutation detection in CRC specimens. ISMD is a highly valuable tool for exploring and understanding the aetiology of ITH and the work is highly relevant to the field.

The paper is well written and sound. I appreciate the combination of experimental techniques and computational modelling, although some decisions about parameters etc. seem to be taken in an arbitrary way. I only have minor comments on the paper.

* it is unclear from the discussion, how SC NAs would influence BaseScope readouts; clearly in the case of a loss signal would be lost also, but what about gains prior/after to mutation? Does that influence the readouts? Maybe it would be worth adding a sentence to the discussion section.

* p.8/l.288: what is the motivation behind the cutoff of 0.05?

* p.9 but also throughout the ms: sometimes percentages and number of cell lines are given, sometimes also the total number of cells. It would be good to make clear what n is in each case and give the actual cell numbers where they are missing

* p.12/l.442: initially?

* p.13/l.501: BaseScope is typically camel cased throughout the ms, not here

* p.13/l.510: its sensitivity

Reviewer #3 (Remarks to the Author):

The manuscript by Baker et al. describes an upgrade (improvement) of a technology, the BaseScope assay by Advanced Cell Diagnostics. The authors claim they improved the sensitivity of the method at the signal amplification system now using one single ZZ pair instead of 20. The advantage of this is the use of short probes and thus the detection of single nucleotide base differences, such as mutant RNAs. They use this method to investigate the sub-clonal architecture in CRC by looking at the spatial distribution of the expression of the driver oncogenes KRAS, BRAF and PIK3CA. From these data, they generate a model showing how time of subclone occurrence and individual subclone fitness influence subclonal mixture within a tumor.

In general, the approach is attractive as it circumvents the necessity to infer expression from DNA sequencing data, it goes beyond this and analyzes differential expression of the same oncogene harboring different SNVs.

The authors use a well-suited cohort of colorectal cancer samples, FFPE material, some of the samples even collected years ago, making them a challenging input into the analysis due to the degradation of RNA/DNA often seen in such samples. Furthermore, the authors try to benchmark their method using cell lines with known mutations in the three oncogenes and demonstrate a high degree of specificity of the method on these samples.

There are several interesting outcomes of their approach, which at the same time raise several questions and concerns.

Major comments:

The technology described is not novel, it has only been improved-on in this publication. The authors claim that they are more sensitive as RNAscope. However, in their validation the authors do not show robustness of the assay when it is tested in vitro on cell lines. A major critical point is that it remains unclear whether the large differences they detect using the BaseScope probes for six KRAS G12X mutations in the cell lines derive from heterogeneous expression or differential sensitivity of the probes. If the latter is the case, their model is not valid. In the discussion, they comment on this observation, indicating themselves that their sensitivity might not be sufficient:

' line 509: Third, whilst the method has high specificity, its sensitivity for detecting heterozygous mutations is typically approximately 25%...'

At the same time, in the introduction and body they state that this technology can be used to provide

new insights into the mechanism of tumor evolution with potential ramifications for selecting patients for treatment. This is therefore a statement, which is not proven in the publication and requires extensive validation using other methods.

In addition, it has been shown by other scientists that mutant KRAS undergoes copy number imbalance often providing a reason for molecular heterogeneity (see PMID:27743339; PMID:23599154). The authors would need to test first on the cell lines, later in tissue samples, whether the differential sensitivity is influenced by CNV or aneuploidy involving the KRAS (and BRAF, PIK3CA) locus.

Upon investigating the subclonal architecture, the authors go slightly beyond the KRAS/BRAF/PIK3CA analysis and compare KI67 proliferation marker expression as well as CD8 cytotoxic T cell infiltration among mutant and wild-type areas. It is not clear, why the authors chose these two additional "markers". Not surprisingly, there is no difference in KI67 between KRAS^{wt} and KRAS^{mut} subclones recapitulating what has been known before – KRAS does not necessarily increase proliferation. The only BRAFV600E tested tumor shows increased KI67, would this be the same if they test another 5? The whole analysis seems random and both R² and p-values are not convincing. If the authors really want to conclude something on differences between wt and mutant clones, they would need to perform a much broader analysis. This might also help to solve the problem with modeling the mixing (see next comment).

As far as the computational model is concerned, I wonder whether the differential expression the authors observed with the different mutants could act as assumed fitness parameters of the clones resulting in a differential mixing score? In addition, the description of the outcome of the mixing simulation to me differentiates between results that are actually not distinguishable (see page 12 upper paragraph). A low mixing score can be obtained either after intermediate appearance and low to intermediate fitness or after late appearance and low to intermediate fitness. This indicates that the model is somewhat superficial in that "fitness" alone might not explain why the same result can be achieved from different starting positions. This is also reflected by the "trend" of correlation (R²=0.24) between the mixing score and the proportion the mutant subclone occupies within the lesion analysed (Suppl. Fig. 7).

Minor comments

The BaseScope result in cell lines and (Fig.1; Suppl. Fig.1), the testing on the tissues (Fig.2) is nicely visible. The BaseScope results for the 3D reconstruction are invisible; resolution/magnification needs to be improved.

Baker et al (NCOMMS-17-11945) Response to Reviewers' comments:

Reviewer #1 (Remarks to the Author):

1. In the present manuscript Baker et al. introduce an improved technique of RNA in situ hybridization. The validation of the technology seems robust. Using this technique the authors continue to identify sub-clonal mutations, which potentially could be missed by bulk sequencing with limited depth. The technique can be an important tool to study intra-tumor heterogeneity, a possible cause of resistance to drug therapy. In short, the method is well demonstrated. Unfortunately the usefulness of its application beyond detection of small clones is not clear.

We are glad that the reviewer considers the BaseScope technique as a potentially important tool. In our opinion the particular value of the technique is that it provides, for the first time, a robust method for spatially resolved mutation detection. Current genetic analysis techniques either do not provide any spatial information whatsoever (e.g. sequencing) or have very limited genetic resolution (e.g. FISH). BaseScope, uniquely, is a very robust methodology that provides single nucleotide resolution whilst also providing spatial data.

2. To facilitate the implication of the technique by others, it would be helpful if the authors explain in some more detail how the technique differs from BaseScope and how the reduction to single ZZ base pairs is achieved.

We thank the reviewer for highlighting this lack of clarity, and we have added text (page 3, paragraph 2, line 98) to better describe that the difference between conventional RNAscope and BaseScope is both a reduction in the number of 1ZZ probes and in the addition of one further signal amplification step.

3. A limitation of the introduced technique is that for each specific (point) mutation, a specific probe is required. It therefore appears that NG bulk sequencing to identify mutations needs to precede RNA ISH, or ISH needs to be performed with a large set of point mutation specific RNA probes. How do the authors envision that this technique can best be applied on naive tumors?

[REDACTED]

4. In addition, how does this technique deal with sampling errors as only a very limited proportion of the tumor can be analyzed?

We note that limited sampling of a tumor is a caveat of all mutation detection methods, and BaseScope is no different in this respect. The principal advantages of BaseScope are:

(a) the sensitivity of detection – we have shown that as few as 20 mutant cells within a field of wild-type cells can be detected – this would certainly be below the limit of detection of standard NGS.

(b) Spatial resolution of mutation location. These spatial data reveal nuances of the evolutionary dynamics of a tumor that are not accessible from frequency information alone. From a clinical perspective, spatial information allows mutations to be assigned to the benign or malignant components of a neoplasm, which may have implications for treatment decisions.

(c) BaseScope can be readily combined with conventional IHC (or FISH) on the same section, providing concurrent analysis of mutation status together with phenotypic information, at cellular resolution.

5. One major challenge in studying intra-tumor heterogeneity is identifying the relative fitness and emergence of a clone from resected material. The problem is that from resected data only the final clone size can be observed, while the fitness advantage, the time at which the clone emerged and random fluctuations all contribute to the final clone size. The authors aim to address this problem, as well as clonal mixing, by using a computational model. I have several more detailed questions regarding this part of the manuscript:

The reviewer highlights the issues inherent to measuring tumor evolutionary dynamics when frequency data from only a single time point is available. One of the main reasons that we are particularly excited about BaseScope technology is that it offers, for the first time, a robust methodology to obtain spatial information in addition to frequency data. Indeed, the key hypothesis of the modeling aspect of our manuscript was that the addition of this topographical information on clone location would help to inform on the underlying evolutionary dynamics, because the dynamics are dependent on spatial structure. Moreover, BaseScope allows clonal populations to be identified *in situ*, so that their microenvironment can be concurrently measured.

6. Do the authors suggest that they can distinguish between the timing and the relative fitness of clones? While this could potentially be very interesting, the manuscript is not clear on this point. As a key observable the authors use clonal mixing. Can they explain how mutational fitness and timing of a clone differ in the pattern of clonal mixing that arises?

We apologize for the lack of clarity here. Whilst we cannot precisely distinguish between the 'age' of a clone and its relative fitness, our mathematical analysis shows that the addition of spatial data (e.g. that accessible via BaseScope) provides additional benefit in resolving these interlinked parameters. Specifically, the posterior distribution of inferred clone ages and relative fitnesses is more constrained (see figure below). Moreover, spatial mixing is not a simple linear correlation with mutant frequency (Supplementary Figure 7c and Supplementary Figure 8a), illustrating how spatial information gives additional information on the evolutionary dynamics.

7. How sensitive are the results to the details of the model?

The sensitivity of the results to the assumptions of the model is an important point that we thank the reviewer for raising.

A key assumption of the model (that determines spatial patterning) is how cells grow and potentially replace existing cells. In our original manuscript, daughter cells would take an empty neighboring position around their parent, or if there was no empty space would create space by pushing the population outwards in a random direction. To determine how pushing determines the spatial dynamics, we have now investigated an alternative model where cells will divide without pushing, and only fill neighboring empty spaces. Spaces are vacated because of cell death (death was not considered in the original model formulation). In this alternative regime, cells divide less frequently in the center (where there tends to be little free space, especially when the cell death is small) compared to the ‘leading edge’.

Our new analysis shows that the qualitative patterns of clone spread and mixing are the same in both these formulations of the model. Specifically, under neutral or near-neutral selection, mixing and mutant frequency are low irrespective of when the tumor is sampled. When selection intensity is strong (mutant has high relative fitness), we see a temporal transit from low mixing to high mixing and then to low mixing again (as the clone is born, expands, and eventually sweeps to near-fixation). We note that the different formulations do lead to quantitative differences. For example, a high death rate produces more empty spaces in the population, and so a mutant with a particular

fitness advantage is more likely to reach fixation than in the ‘pushing model’ case (see figure below).

8. Why is clonal mixing being used? What is the advantage over the mutation frequency? What is the relation between these two observables in the model? And how would changes in adhesion or migration (which are not captured in the model) affect the clonal mixing, given a certain mutation frequency?

We investigated clonal mixing for two reasons: first, because spatial resolution of mutant clones is the novel measurement provided by the BaseScope method, and second because we hypothesized that spatial information would provide additional resolution on the evolutionary dynamics, over-and-above frequency information alone. This latter point was motivated from experience in classical evolutionary biology experiments, as is noted above.

Our experimental and simulation data both show that mutant frequency and clonal mixing are not linearly correlated, but in fact show a complex relationship (new Supplementary Figure 8a, reproduced below).

We agree with the reviewer there are many other factors that potentially influence clonal mixing, especially during metastasis, that we have not considered (principally cell migration), that will be extremely interesting to study in future work. Indeed, we envisage that the spatial resolution provided by BaseScope will be an important tool for

helping to resolve such dynamics. For now, we note that the analysis in this manuscript is restricted to primary tumors where migration is likely fairly minimal (indeed the observed mutant clones are contiguous).

9. The dynamics of cells in their simulation seems unrealistic. Adenocarcinoma's are thought to at least partially grow by a process of gland fission, not only random movement of cells. The assumed dynamics probably strongly influences the clonal mixing. What is the motivation for the chosen model/dynamics?

We note that our experimental data showed that, typically, each gland in the colorectal tumors was *clonal* – e.g. it contained only mutant or non-mutant cells, but not both. As the reviewer notes, tumor growth is driven by gland rather than cell division *per se*, and consequently we considered that it was reasonable to equate cells in our model to glands in a primary tumor.

10. In Fig. 4 the results are plotted against time. Is it the relative time at which a clone occurs or the absolute time, i.e. the age of a tumor that is studied here? If the latter is the case, can the authors translate the inferred value to an actual age of the tumor?

The time in Figure 4 is the time when the tumor is sampled (e.g. tumor age). We are reluctant to attempt to transform this simulated time to provide an estimate of actual elapsed time in tumor growth, because of the level of abstraction in the model and the uncertainty around parameter values (principally cell division rates). We consider that the analysis as-is well illustrates how the level of clonal mixing evolves over time, and can potentially be used to constrain inferences of evolutionary dynamics.

Reviewer #2 (Remarks to the Author):

The paper by Baker et al. describes the application of the BaseScope assay for in-situ mutation detection in CRC specimens. ISMD is a highly valuable tool for exploring and understanding the aetiology of ITH and the work is highly relevant to the field.

The paper is well written and sound. I appreciate the combination of experimental techniques and computational modelling, although some decisions about parameters etc. seem to be taken in an arbitrary way. I only have minor comments on the paper.

We are grateful for this positive assessment of our manuscript.

We note that a detailed discussion about the model construction and parameter values is provided to the points 5-10 of Reviewer 1's critique above.

1. it is unclear from the discussion, how SCNAs would influence BaseScope readouts; clearly in the case of a loss signal would be lost also, but what about gains prior/after to mutation? Does that influence the readouts? Maybe it would be worth adding a sentence to the discussion section.

We agree that SCNAs are likely to influence mRNA expression levels, and therefore may also influence the BaseScope readout of a WT or mutant transcript (this point was also raised by Reviewer #3 point 3). To investigate further, we have analyzed the relationship between *KRAS* mRNA expression and *KRAS* copy number status in cell lines (n=877, Cancer Cell Line Encyclopedia¹) and colorectal cancers (n = 421, TCGA²).

As expected, there is a trend between increased *KRAS* copy number and elevated mRNA expression, therefore in general we would expect a gain of a mutant allele to increase the probability of detection by BaseScope, and a loss of that allele to decrease that probability. We have added a sentence to our discussion to highlight this. We note that there are other methods widely in use (e.g. FISH) for the *in situ* analysis of SCNAs, BaseScope however is the first robust method for the *in situ* detection of point

mutations, and our analysis was agnostic of CN state.

2. p.8/l.288: what is the motivation behind the cutoff of 0.05?

The threshold of $\epsilon=0.05$ was arbitrarily chosen as sufficiently low threshold in our Approximate Bayesian Computation (ABC) algorithm to ensure that accepted simulation parameters produced output that sufficiently resembled the experimental data. Smaller values of ϵ of course would lead to a more accurate fit of the model (smaller estimated parameter range), but at the expense of requiring substantially more simulations. Below we provide a plot of ϵ versus the standard deviation of the marginal posterior distributions for sampling time and relative fitness of the mutant, demonstrating that $\epsilon=0.05$ was a reasonable cutoff to choose. The size of estimated parameter range has a relatively small change with $\epsilon < 0.1$.

3. p.9 but also throughout the ms: sometimes percentages and number of cell lines are given, sometimes also the total number of cells. It would be good to make clear what n is in each case and give the actual cell numbers where they are missing

We apologize for the lack of consistency in the reporting of data, and we have now ensured that in all cases we have provided the number of cells lines/tumors counted and percentages. Absolute cell numbers that were counted can be found in Supplementary Table 4.

4. p.12/l.442: initially?

5. p.13/l.501: BaseScope is typically camel cased throughout the ms, not here

6. p.13/l.510: its sensitivity

We thank the reviewer for the thorough analysis of the text, and have corrected these errors accordingly.

Reviewer #3 (Remarks to the Author):

The manuscript by Baker et al. describes an upgrade (improvement) of a technology, the BaseScope assay by Advanced Cell Diagnostics. The authors claim they improved the sensitivity of the method at the signal amplification system now using one single ZZ pair instead of 20. The advantage of this is the use of short probes and thus the detection of single nucleotide base differences, such as mutant RNAs. They use this method to investigate the sub-clonal architecture in CRC by looking at the spatial distribution of the expression of the driver oncogenes KRAS, BRAF and PIK3CA. From these data, they generate a model showing how time of subclone occurrence and individual subclone fitness influence subclonal mixture within a tumor.

In general, the approach is attractive as it circumvents the necessity to infer expression from DNA sequencing data, it goes beyond this and analyzes differential expression of the same oncogene harboring different SNVs.

The authors use a well-suited cohort of colorectal cancer samples, FFPE material, some of the samples even collected years ago, making them a challenging input into the analysis due to the degradation of RNA/DNA often seen in such samples. Furthermore, the authors try to benchmark their method using cell lines with known mutations in the three oncogenes and demonstrate a high degree of specificity of the method on these samples.

We thank the reviewer for this accurate summary of our manuscript.

There are several interesting outcomes of their approach, which at the same time raise several questions and concerns.

Major comments:

1. The technology described is not novel, it has only been improved-on in this publication. The authors claim that they are more sensitive as RNAscope.

BaseScope constitutes a major advance of the technology first developed for RNAscope. We reduced the number of ZZ probes used for signal detection from ~20 probes to a single ZZ pair, and to provide adequate sensitivity we then had to develop an extra layer of signal amplification. As we demonstrate in the manuscript, BaseScope is able to detect single nucleotide variants *in situ* – this level of sequence discrimination was not possible with RNAscope, and hence BaseScope is a more sensitive assay.

2. However, in their validation the authors do not show robustness of the assay when it is tested *in vitro* on cell lines.

We note that our testing on 9 pairs of mutant and wild-type cell lines (Figure 1b, Figure S1a-g) shows perfect specificity of the BaseScope assay when it is used to probe a

single point mutation at a time. Furthermore we have shown sufficient sensitivity to always detect a true-positive signal in all confirmed mutant cell pellets (n=9) and tumors (n=30) that were tested – i.e. BaseScope did not fail to detect a confirmed mutation. We would argue that these data alone give a persuasive demonstration of the robustness of the technique.

If the reviewer intended to question the reproducibility of the assay, shown below are further data representing 5 *KRAS* mutant cell lines or tumors that were tested twice with the BaseScope assay (with >3 months separating the two repeats). We show the signal is very reproducible, with no significant difference in signal quantification in any of the samples. These reproducibility data have been added the manuscript (new Supplementary Figure 2b, also below).

3. A major critical point is that it remains unclear whether the large differences they detect using the BaseScope probes for six *KRAS* G12X mutations in the cell lines derive from heterogeneous expression or differential sensitivity of the probes. If the latter is the case, their model is not valid. In the discussion, they comment on this observation, indicating themselves that their sensitivity might not be sufficient:

‘ line 509: Third, whilst the method has high specificity, its sensitivity for detecting heterozygous mutations is typically approximately 25%...’

The reviewer is questioning whether heterogeneous expression (e.g. ‘biological noise’) or differential sensitivity (e.g. ‘technical noise’) impacts the BaseScope signal. The answer is of course that both influence the strength of the observed signal.

To quantify biological noise, we investigated the heterogeneity of *KRAS* mRNA expression in the presence of the a *KRAS* G12X mutation in cell lines (n=877, Cancer Cell Line Encyclopedia¹) and colorectal cancers (n=421, TCGA²) of known mutational status.

Both cell lines and tumors showed large variability in *KRAS* mRNA expression, irrespective of which *KRAS* mutation was present, as shown below.

To further emphasize this point, we show below highly significant difference in WT *KRAS* expression between two cell lines, as measured by BaseScope signal. Biological noise is therefore a considerable factor even in cell line pellets (where we can reasonably expect RNA quality to be equivalent between pellets). We note that the inherent variation in RNA quality between archival FFPE samples will further increase the observed variability in BaseScope signal.

Heterogeneity of WT *KRAS* expression by BaseScope

Quantifying the technical noise is extremely challenging, and in our opinion such a quantification is unnecessary. The inherent biological noise described above means that it is not possible to deduce the absolute sensitivity of a probe from the analysis of BaseScope readouts in cell lines or tumors, and so it is for this reason that we have not claimed BaseScope signal to be a proxy for absolute or relative mRNA expression levels. While there is undoubtedly variation in probe sensitivity, we reiterate that our data show that in cell lines (Figure 1b and Supplementary Figure 1a-g) that BaseScope always detects the point mutation that is known to be in the sample, typically in the majority of cells. Moreover, the level of detected signal is consistent between technical replicates, as shown above.

It is important to note that different sensitivities of probes do not invalidate our fundamental findings. The glandular organization of colorectal cancers means that we reliably assign a gland as mutant if we observe BaseScope signal above the level of non-specific binding (approx. 1 in 400 cells). As we describe in the manuscript, probe sensitivity in FFPE tumors is variable, however the typical detection of BaseScope signal within 1 in 4 mutant cells is by far sufficient to confidently assign a tumor gland (typically containing tens or hundreds of cells in a single section) as a true mutant.

4. At the same time, in the introduction and body they state that this technology can be used to provide new insights into the mechanism of tumor evolution with potential ramification for selecting patients for treatment. This is therefore a statement, which is not proven in the publication and requires extensive validation using other methods.

There is extensive literature in the evolutionary ecology field that makes use of the spatial distribution of individuals in a population to examine how spatial structure constrains population evolutionary dynamics (see for example: doi:10.1098/rspb.2007.0529) and to examine the relationship between microenvironmental factors and population evolution (see for example the review: doi:10.1016/S0169-5347(03)00008-9). The broad class of tools used for these analyses are classified as phylogeography or landscape genetics methods. Cancer follows the same evolutionary principles as other evolving systems, and so we should expect spatial information to be of similar value in cancer and classical evolutionary biology alike.

To provide a quantitative demonstration that spatial information improves the accuracy with which evolutionary dynamics can be inferred, we consider the posterior distributions for the inferred selective advantage and time of appearance of subclones that were inferred using (a) only frequency data, or (b) the combination of frequency and spatial data (e.g. mixing scores). The posterior distributions were narrower and more closely resembled the truth when spatial data were included (See the figure in the response to Reviewer 1 point 6).

With respect to the clinical utility of BaseScope, upon detailed analysis of adenomas with foci of cancer ('Ca-in-ad' samples), we identified two cases wherein a *KRAS* mutation was localized to the benign adenomatous region of the lesion, with the invasive cancer exclusively *KRAS* wild-type. This information would not be accessible by standard next generation sequencing analysis, and may render such patients eligible for targeted therapy that is usually reserved for *KRAS* wild-type tumors, and we have now highlighted this important point in the manuscript. However, we agree that our manuscript does not prove that BaseScope aids decision-making for treatment selection, and have tempered this statement appropriately ("The BaseScope assay represents a significant technical advance for *in situ* mutation detection that provides new insight into the mechanisms of tumor evolution, and **could have** ramifications for selecting patients for treatment"). Furthermore, we have added a note to the discussion to highlight that additional work is required to investigate the clinical utility of the

method.

5. In addition, it has been shown by other scientists that mutant KRAS undergoes copy number imbalance often providing a reason for molecular heterogeneity (see PMID:27743339; PMID:23599154). The authors would need to test first on the cell lines, later in tissue samples, whether the differential sensitivity is influenced by CNV or aneuploidy involving the KRAS (and BRAF, PIK3CA) locus.

We agree that copy number status is a factor that can influence gene expression level, and consequently influence BaseScope signal – please see our response to Reviewer 2, point 1. We find that there is a correlation between increased *KRAS* copy number state and *KRAS* mRNA expression, but there is much variation and overlap between the copy number groups. Therefore we conclude that a gain in copy number is likely to increase the probability of detection by BaseScope, whereas a copy number loss decreases it. We have added a sentence to our discussion to highlight this.

Due to the variable and sample-dependent influence of biological factors and RNA degradation in FFPE blocks (discussed above in response to point 3), it is therefore not possible to infer to what extent copy number alterations are responsible for the differing sensitivity of the BaseScope readout. However we note that our analysis is agnostic of CN state - we detect mutant subclones by the presence or absence of BaseScope signal in a tumor gland, not by the absolute quantification of that signal. Moreover, we reiterate that the novelty of the BaseScope assay is for *in situ* detection of point mutations, and that our data show a very high overall sensitivity of point mutation detection.

6. Upon investigating the subclonal architecture, the authors go slightly beyond the KRAS/BRAF/PIK3CA analysis and compare KI67 proliferation marker expression as well as CD8 cytotoxic T cell infiltration among mutant and wild-type areas. It is not clear, why the authors chose these two additional “markers”. Not surprisingly, there is not difference in KI67 between KRASwt and KRASmut subclones recapitulating what has been known before – KRAS does not necessarily increase proliferation. The only BRAFV600E tested tumor shows increased KI67, would this be the same if they test another 5? The whole analysis seems random and both R2 and p-values are not convincing. If the authors really want to conclude something on differences between wt and mutant clones, they would need to perform a much broader analysis. This might also help to solve the problem with modeling the mixing (see next comment).

The reviewer is correct to highlight that these supplementary data are illustrative only. We used KI67 and CD8 to illustrate the potential of combining phenotypic analysis (cell proliferation and immunogenicity) with BaseScope, not to draw any significant biological conclusions. We consider that the exciting potential combination of BaseScope and IHC will be to facilitate *in situ* phenotyping of tumor subclones in human tumors, and indeed a complete phenotypic investigation of tumor subclones will be the subject of our

further work.

7. As far as the computational model is concerned, I wonder whether the differential expression the authors observed with the different mutants could act as assumed fitness parameters of the clones resulting in a differential mixing score? In addition, the description of the outcome of the mixing simulation to me differentiates between results that are actually not distinguishable (see page 12 upper paragraph). A low mixing score can be obtained either after intermediate appearance and low to intermediate fitness or after late appearance and low to intermediate fitness. This indicates that the model is somewhat superficial in that “fitness” alone might not explain why the same result can be achieved from different starting positions. This is also reflected by the “trend” of correlation ($R^2=0.24$) between the mixing score and the proportion the mutant subclone occupies within the lesion analysed (Suppl. Fig. 7).

The fitness in our model is defined as the net growth rate of a clone. Our immunohistochemical analysis typically showed no relationship between the proliferative index (Ki67) and mutation status of a clone (Supplementary Figure 6b), and as we have explained above we are not able to measure absolute expression levels using BaseScope. Consequently, we do not think that clone fitness can be simply inferred by the measured level of mutant proto-oncogene expression.

Following the reviewers’ comments we have performed a more extensive investigation of the relationship between mutant clone frequency and mixing score. The original regression on the empirical data showed poor evidence of a correlation (low R^2), and further analysis of our computational model confirms that there is a complex non-linear relationship between the two measures (see new Supplementary Figure 8a, and also our reply to reviewer 1 point 8). Consequently we have removed the linear fit line in Supplementary Figure 7c.

We agree with the reviewer that the arising time and the fitness are entangled. It was not our intention to suggest that either clone fitness or arising time alone could explain the mixing level and mutant frequency observed in samples, and we apologize for the lack of clarity if this was understood. Instead, our model illustrates how the combination of the two evolutionary parameters can lead to similar mixing levels and mutant frequencies but with different temporal trajectories and at different probabilities (Supplementary Figure 8). We used Approximate Bayesian Computation to infer the parameter combinations that had the highest probabilities of recapitulating the observed empirical data in Figure 4 of the main text.

Minor comments

8. The BaseScope result in cell lines and (Fig.1; Suppl. Fig.1), the testing on the tissues (Fig.2) is nicely visible. The BaseScope results for the 3D reconstruction are invisible; resolution/magnification needs to be improved.

We apologize for the poor resolution of the BaseScope signal in Supplementary Figure 5. We have replaced the image with a high-resolution copy, and also provided a high magnification inset.

References - Response to Reviewer's comments

- 1 Barretina, J. *et al.* The Cancer Cell Line Encyclopedia enables predictive modelling of anticancer drug sensitivity. *Nature* **483**, 603-607, doi:10.1038/nature11003 (2012).
- 2 Cancer Genome Atlas, N. Comprehensive molecular characterization of human colon and rectal cancer. *Nature* **487**, 330-337, doi:10.1038/nature11252 (2012).

REVIEWERS' COMMENTS:

Reviewer #1 (Remarks to the Author):

All my concerns have been addressed. I appreciate the thorough responses and congratulate the authors with their fine work.

Reviewer #2 (Remarks to the Author):

The authors addressed all remarks to my satisfaction. Thanks for updating the ms.

Reviewer #3 (Remarks to the Author):

I appreciate the authors efforts to improve the manuscript. My comments were answered in a satisfying manner and I have no objections against publication of this manuscript.

The only point I still disagree is the KI67, C D8 affair, which is done and presented in a superficial manner. Although I understand the authors' reply, I am wondering whether they really need this part as it does not add much to their conclusions.

We are grateful to all three reviewers, whose critiques have helped us to improve our manuscript. We are pleased that all three reviewers consider that we have fully addressed their concerns, and we will be happy to see our correspondence with the reviewers published alongside our manuscript.

Reviewer #1 (Remarks to the Author):

All my concerns have been addressed. I appreciate the thorough responses and congratulate the authors with their fine work.

Reviewer #2 (Remarks to the Author):

The authors addressed all remarks to my satisfaction. Thanks for updating the ms.

Reviewer #3 (Remarks to the Author):

I appreciate the authors efforts to improve the manuscript. My comments were answered in a satisfying manner and I have no objections against publication of this manuscript.

The only point I still disagree is the KI67, CD8 affair, which is done and presented in a superficial manner. Although I understand the authors' reply, I am wondering whether they really need this part as it does not add much to their conclusions.

We appreciate the reviewer's concern, but nevertheless would prefer to keep the immunophenotyping analysis in our manuscript (e.g. to retain Fig S6). We feel these data have value for the narrative of our manuscript for two reasons. First because they demonstrate that the BaseScope assay can be readily combined with immunohistochemistry, and second because they provide, we think for the first time, some *in situ* phenotypic information on *KRAS*, *BRAF* and *PIK3CA*- mutant tumour subclones in colorectal cancer.